# The DEformer: An Order-Agnostic Distribution Estimating Transformer

Michael A. Alcorn [1]  Anh Nguyen [1]

## Abstract

Order-agnostic autoregressive distribution (density) estimation (OADE), i.e., autoregressive distribution estimation where the features can occur in an arbitrary order, is a challenging problem in generative machine learning. Prior work on OADE has encoded feature identity by assigning each feature to a distinct fixed position in an input vector. As a result, architectures built for these inputs must strategically mask either the input or model weights to learn the various conditional distributions necessary for inferring the full joint distribution of the dataset in an order-agnostic way. In this paper, we propose an alternative approach for encoding feature identities, where each feature's identity is included *alongside* its value in the input. This feature identity encoding strategy allows neural architectures designed for sequential data to be applied to the OADE task without modification. As a proof of concept, we show that a Transformer trained on this input (which we refer to as "the DEformer"[2], i.e., the distribution estimating Transformer) can effectively model binarized-MNIST, approaching the performance of fixed-order autoregressive distribution estimating algorithms while still being entirely order-agnostic. Additionally, we find that the DEformer surpasses the performance of recent flow-based architectures when modeling a tabular dataset.

## 1. Introduction

For tasks such as: (a) efficiently imputing arbitrary missing values from an input or (b) preemptive anomaly detection in systems where input features can arrive asynchronously in an arbitrary order (e.g., internet of things applications (Ahmad et al., 2017)), order-agnostic autoregressive distribution (density) estimation (OADE) is necessary. However, because there are $D!$ factorizations of the joint probability for a $D$-dimensional input, order-agnosticism adds considerable complexity to the distribution estimation task. As a result, many likelihood-based generative models either: (1) assume a single, fixed order for the input features (e.g., NADE (Larochelle & Murray, 2011), PixelRNN (Oord et al., 2016), and TraDE (Fakoor et al., 2020)), (2) only use a small subset of the possible feature orderings in practice (e.g., MADE (Germain et al., 2015), IAF (Kingma et al., 2016), MAF (Papamakarios et al., 2017), and LMConv (Jain et al., 2020)), or (3) are not autoregressive (e.g., some flows (Dinh et al., 2015; 2017; Kingma & Dhariwal, 2018; Papamakarios et al., 2021)).

In contrast to the previously mentioned approaches, Deep-NADE (Uria et al., 2014; 2016) is notable in that it performs full OADE. Specifically, DeepNADE consists of a standard multilayer perceptron (MLP) that takes as input the concatenation of a $D$-dimensional binary mask $\mathbf{m}$ and the masked version of the sample $\mathbf{x}$, $\hat{\mathbf{x}} = \mathbf{m} \odot \mathbf{x}$, i.e., the input $[\hat{\mathbf{x}}, \mathbf{m}]$ is a vector of size $2D$. The feature identities (e.g., pixel locations) are thus encoded by their positions in the input feature vectors. However, this input design precludes the use of neural architectures that are designed for sequential data (e.g., recurrent neural networks and Transformers (Vaswani et al., 2017))—models that are a natural fit for autoregressive problems.

Taking inspiration from a recently described multi-agent spatiotemporal Transformer (Alcorn & Nguyen, 2021b), in this paper, we propose an alternative approach for encoding feature identities, where each feature's identity is included *alongside* its value in the input. Using this input design, we train an otherwise ordinary Transformer (which we refer to as "the DEformer", i.e., the distribution estimating Transformer) to perform OADE on the binarized-MNIST (Salakhutdinov & Murray, 2008) and POWER (Vergara et al., 2012) datasets. We find that:

1. The DEformer—while being entirely order agnostic and autoregressive—is competitive with fixed-order distribution estimating algorithms when modeling binarized-MNIST and surpasses recent flow-based ar-

---

[1]Department of Computer Science and Software Engineering, Auburn University, Auburn, Alabama, USA. Correspondence to: Michael A. Alcorn <alcorma@auburn.edu>.

Third workshop on *Invertible Neural Networks, Normalizing Flows, and Explicit Likelihood Models* (ICML 2021). Copyright 2021 by the author(s).

[2]All data and code for the paper are available at: https://github.com/airalcorn2/deformer.

chitectures when modeling the tabular POWER dataset.

2. The DEformer can effortlessly fill in pixels of binarized-MNIST images that are missing in a variety of patterns.

3. The DEformer can easily distinguish between binarized-MNIST and non-binarized-MNIST images.

## 2. Architecture

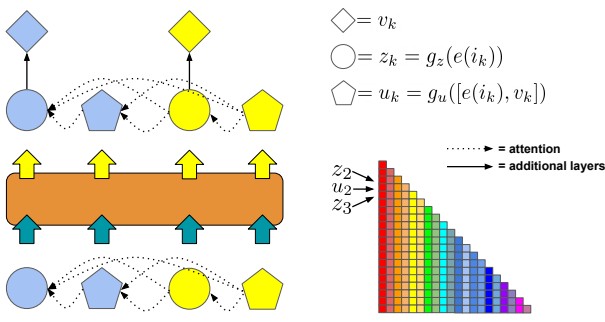

*Figure 1.* By including each feature's identity *alongside* its value in the input, sequential models can be used to perform order-agnostic autoregressive distribution estimation. The DEformer is a Transformer that uses an interleaved input design (partially depicted here with the self-attention mask) for this task. The two sets of interleaved feature vectors consist of identity feature vectors ($z_k$) and identity/value feature vectors ($u_k$), and $g_z$ and $g_u$ are their respective multilayer perceptrons. For the binarized-MNIST dataset, each feature identity $i_k$ is a tuple ($r_k, c_k$) where $r_k$ and $c_k$ are the row and column for the pixel indexed by $k$ in the permuted sequence, respectively, and $v_k$ is the value of the pixel (which is zero or one for binary images). For tabular data, each $i_k$ corresponds to a column, and $v_k$ is the value of the indexed column in the row. Lastly, $e$ is an identity encoding function, which is simply the identify function in the case of binarized-MNIST, and is an embedding layer for tabular data.

Here, we describe our order-agnostic distribution estimating Transformer, the DEformer (Figure 1). The goal in OADE is to model the joint distribution of a $D$-dimensional vector **x** by exploiting the chain rule of probability, i.e.:

$$p(\mathbf{x}) = \Pi_{d=1}^{D} p(x_{o_d} | \mathbf{x}_{o_{<d}})$$

where, as in Uria et al. (2014), $o$ is a $D$-tuple representing a permutation of the elements in **x**, so $x_{o_d}$ indicates the element of **x** indexed by the $d$-th element of $o$, and $\mathbf{x}_{o_{<d}}$ means the elements in **x** indexed by the first $d - 1$ elements of $o$. We assume each discrete feature can take on one of $C$ labels (which is the case for image datasets), but, in theory, each feature could have a different number of possible labels.

Rather than encoding each feature's identity by confining it to a specific position in the input, here, we propose including the feature's identity as an additional input variable *alongside* its value. Specifically, the input to the DEformer consists of two parallel sequences: one containing only feature identities, and another containing identity/value pairs:

1. $i_1, i_2, ..., i_n$

2. $(i_1, v_1), (i_2, v_2), ..., (i_n, v_n)$

where $i_k$ is the identity of the $k$-th feature in the permuted sequence and $v_k$ is the value of the $k$-th feature. In the case of binarized-MNIST, each $i_k$ is a tuple ($r_k, c_k$) indicating the row and column of the pixel, respectively, and $v_k$ is the value of the pixel (i.e., zero or one). For tabular data, each $i_k$ indexes a column, and $v_k$ is the value of the indexed column in the row.

The identity inputs are mapped to identity feature vectors using an MLP, i.e., $z_k = g_z(e(i_k))$ where $z_k$ is the identity feature vector, $g_z$ is the identity MLP, and $e$ is an identity encoding function. In the case of binarized-MNIST, $e$ is simply the identity function, i.e., $e(i_k) = [r_k, c_k]$, while for tabular data, $e$ is an embedding layer. The identity/value pairs are similarly mapped to identity/value feature vectors using a separate MLP, i.e., $u_k = g_u([e(i_k), v_k])$ where $u_k$ is the identity/value feature vector and $g_u$ is the identity/value MLP. These two sets of feature vectors are interleaved with one another (i.e., $u_k$ always immediately follows $z_k$ in the input) to form a $2D \times F$ matrix where $F$ is the dimension of the outputs for the MLPs.

This matrix is passed into the Transformer along with a lower triangular self-attention mask, which encodes the following dependencies (see Figure 1):

1. When processing $z_{k_2}$, the DEformer is allowed to "look" at: (i) any $z_{k_1}$ where $k_1 \leq k_2$ and (ii) any $u_{k_1}$ where $k_1 < k_2$.

2. When processing $u_{k_2}$, the DEformer is allowed to "look" at: (i) any $z_{k_1}$ where $k_1 \leq k_2$ and (ii) any $u_{k_1}$ where $k_1 \leq k_2$.

Like Alcorn & Nguyen (2021a;b), we do not use positional encoding (Vaswani et al., 2017) because Irie et al. (2019) observed that positional encoding is not only unnecessary, but detrimental for Transformers that use a causal attention mask.

Each processed $z_k$ feature vector is then passed through a final linear layer. When modeling discrete features, the final linear layer is followed by a softmax, which gives a probability distribution over the labels for the feature indexed by $k$. The loss for each sample is thus:

$$\mathcal{L} = \sum_{k=1}^{K} -\ln(f(Z)_{2k-1}[v_k]) \tag{1}$$

where $f(Z)_{2k-1}[v_k]$ is the probability assigned to the label $v_k$ (where $v_k$ is an integer from one to $C$) by $f$, i.e., Equation (1) is the NLL of the data according to the model. For continuous features, the output of the final linear layer defines a mixture of Gaussians, so the loss for each sample is:

$$\mathcal{L} = \sum_{k=1}^{K} -\ln(\pi_k \cdot c_k)$$

where $\pi_k = \text{softmax}(f(Z)_{2k-1,1:J})$ is a vector containing the $J$ mixture proportions for feature $k$, and $c_k$ is a vector containing the mixture densities such that:

$$c_k[j] = \frac{1}{\sigma_{k,j}\sqrt{2\pi}} e^{-\frac{1}{2}(\frac{v_k - \mu_{k,j}}{\sigma_{k,j}})^2}$$

where $\sigma_k = f(Z)_{2k-1,J+1:2J}$ and $\mu_k = f(Z)_{2k-1,2J+1:3J}$.

Because any ordering of a chain rule decomposition of a joint probability produces the same value, e.g.:

$$p(x_1)p(x_2|x_1)p(x_3|x_1x_2) = p(x_3)p(x_2|x_3)p(x_1|x_3x_2)$$

like Uria et al. (2014); Yang et al. (2019); Alcorn & Nguyen (2021b), we shuffle the order of the features in each training sample to encourage the DEformer to learn a joint distribution of the dataset that is approximately permutation invariant with respect to the ordering of the features.

## 3. Experiments

To test the utility of the DEformer for OADE, we trained a nearly identical architecture to the model described in Alcorn & Nguyen (2021b) on the binarized-MNIST (Salakhutdinov & Murray, 2008) and POWER (Vergara et al., 2012) datasets. The binarized-MNIST dataset consists of 70,000 $28 \times 28$ pixel binary images (i.e., the pixel values are either black or white) of digits (i.e., 0-9) where each digit is represented by the same number of images. We used the standard 60,000/10,000 split for training/testing images, respectively, and used 1,200 of the 60,000 training images (i.e., 2%) for validation. The POWER dataset consists of 2,049,280 power measurements from a single household in a tabular format, where each sample consists of six real values. We used the same preprocessing steps and training/validation/test split described in Papamakarios et al. (2017).

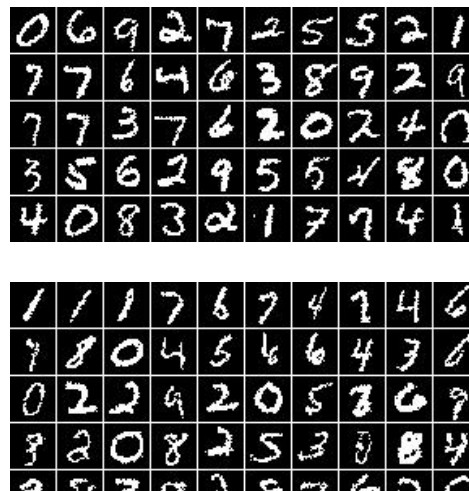

Figure 2. **Top**: A sample of 50 images from the test set of binarized-MNIST organized by their average NLLs according to the DEformer (starting with the lowest average NLL, 42.1, at the top left, and ending with the highest average NLL, 119.6, at the bottom right). **Bottom**: 50 generated images organized by their average NLLs according to the DEformer (starting with the lowest average NLL, 42.0, at the top left, and ending with the highest average NLL, 131.1, at the bottom right). The average NLLs for both sets of images are calculated over 10 random orderings.

The size of the output for the final linear layer was one for the binarized-MNIST dataset and $3 \times 150 = 450$ for the POWER dataset (as in Fakoor et al. (2020)), but all remaining hyperparameters and training details were nearly identical to `baller2vec++` (Alcorn & Nguyen, 2021b), which itself closely follows the original Transformer (Vaswani et al., 2017). Specifically, the Transformer settings were: $d_{\text{model}} = 512$ (the dimension of the input and output of each Transformer layer), eight attention heads, $d_{\text{ff}} = 2048$ (the dimension of the inner feedforward layers), six layers, dropout probabilities of 0.0 and 0.2 for the binarized-MNIST and POWER datasets, respectively, and no positional encoding. Each MLP (i.e., $g_z$, $g_u$, and $g_r$) had 128, 256, and 512 nodes in its three layers, respectively, and a ReLU nonlinearity following each of the first two layers. Lastly, the identity embedding layer for the POWER dataset mapped column indices to 20-dimensional vectors.

We used the Adam optimizer (Kingma & Ba, 2015) with an initial learning rate of $10^{-6}$, $\beta_1 = 0.9$, $\beta_2 = 0.999$, and $\epsilon = 10^{-9}$ to update the model parameters, of which there were ~19 million. The learning rate was reduced to $10^{-7}$ after 5/20 epochs of the validation loss not improving for the binarized-MNIST/POWER datasets, respectively, and we used batch sizes of 1/128 for the binarized-MNIST/POWER datasets, respectively. Models were implemented in PyTorch and trained on a single NVIDIA GTX 1080 Ti GPU for ~50/700 epochs (2.5/6 days) for

*Table 1.* The average NLL on the binarized-MNIST test set for different models. Despite being entirely order agnostic ("OA"), the DEformer is competitive with PixelRNN and TraDE, which use a single fixed order ("FO"). For MADE, the model was trained on 32 different orders (Uria et al., 2016). The average NLLs for both DeepNADE and the DEformer are calculated over 10 random orderings.

| MODEL | NLL |
|---|---|
| DEEPNADE (OA) | 89.17 |
| MADE (32) | 86.64 |
| PIXELRNN (FO) | 79.20 |
| TRADE (FO) | 78.92 |
| DEFORMER (OA) | 80.49 |

*Table 2.* The average NLL on the POWER test set for different models. The DEformer surpasses the performance of recent flow-based architectures while still retaining order-agnostic and autoregressive properties. The average NLL for the DEformer is calculated over 10 random orderings.

| MODEL | NLL |
|---|---|
| REALNVP | -0.17 |
| MAF | -0.3 |
| NAF | -0.62 |
| NSF | -0.66 |
| TRADE | -0.73 |
| DEFORMER | -0.68 |

the binarized-MNIST/POWER datasets, respectively, and the validation set was used for early stopping.

## 4. Results

The DEformer achieved an average NLL (taken over 10 orders) of 80.49 on the binarized-MNIST test set. This is a vast improvement over DeepNADE (Uria et al., 2014) and is competitive with fixed-order distribution estimation algorithms like PixelRNN (Oord et al., 2016) and TraDE (Fakoor et al., 2020) (see Table 1). On the POWER dataset, the DEformer achieved an average NLL of -0.68, which surpasses the performance of recent flow-based architectures like NAF (Huang et al., 2018) and NSF (Durkan et al., 2019). We suspect the DEformer's performance could be improved with a careful hyperparameter search.

Following Uria et al. (2014), Figure 2 shows 50 samples from the test set of binarized-MNIST sorted by their average NLLs (taken over 10 orders) according to the DEformer, along with 50 samples generated by the DEformer, also sorted by their average NLLs. Also following Uria et al. (2014), Figure 3 shows examples of images with 100 pixels missing in a variety of patterns, which were then "filled in" by the DEformer when conditioned on the remaining 684

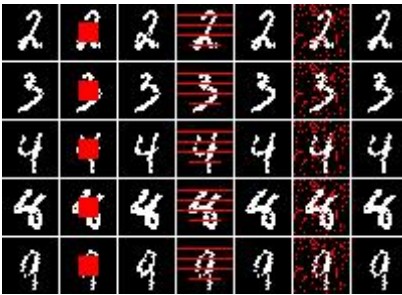

*Figure 3.* Because the DEformer is order-agnostic, it can easily "fill in" images where pixels are missing in a variety of patterns by placing the missing pixels at the end of the input sequence. Here, each row corresponds to a different ground truth image from the test set (depicted in the first column). The remaining pairs of columns show 100 removed pixels (red) from the ground truth image and the corresponding filled in image.

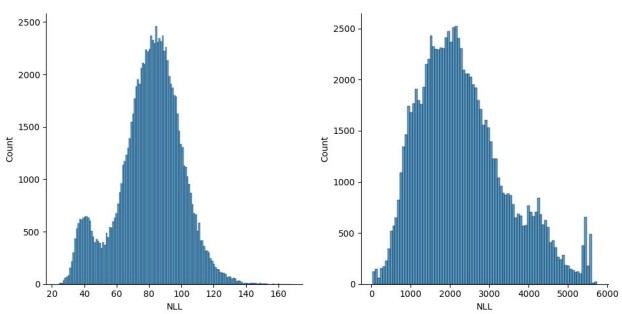

*Figure 4.* The distribution of the DEformer average NLLs for the binarized-MNIST test set (left) and a subset of 10,000 images from the binarized-notMNIST dataset (right) diverge considerably (notice the difference in scales of the $x$-axes), i.e., the DEformer consistently assigns lower probabilities to out-of-distribution samples.

pixels. Like DeepNADE, this task is trivial for the DEformer because the pixels can be arranged such that the conditioning pixels are at the beginning of the sequence. Lastly, as can be seen in Figure 4, the DEformer can easily distinguish between in-distribution and out-of-distribution (i.e., binarized-notMNIST images (Bulatov, 2011)) samples.

## 5. Related Work

### 5.1. Interleaved input Transformers

The DEformer is directly inspired by `baller2vec++` (Alcorn & Nguyen, 2021b), a multi-agent spatiotemporal Transformer that used an identical interleaved input design to model the behaviors of coordinated agents. Our key contribution is recognizing that this interleaved architecture design can be applied to OADE. The DEformer is architecturally similar to the independently developed XLNet language model (Yang et al., 2019). Compared to XLNet, the DEformer:

1. encodes feature identity by including it as an input to the network (instead of using positional embeddings) and

2. uses a full lower triangular attention mask to attend to *both* identity feature vectors and identity/value feature vectors that occur earlier in the shuffled input (instead of only attending to the "content stream").

Notably, XLNet was trained to only predict the final six tokens of a shuffled sentence because the authors observed "slow convergence in preliminary experiments". The DEformer was capable of modeling the values for all 784 pixels in our binarized-MNIST experiments.

### 5.2. DeepNADE

One important way DeepNADE (Uria et al., 2014; 2016) and the DEformer differ is in the size of the outputs for their final classification layers, which are $DC$ and $C$, respectively. While this difference is not particularly important for a relatively simple dataset like binarized-MNIST, for more complex datasets like CIFAR-10 (Krizhevsky et al., 2009), these contrasting designs produce dramatically different parameter counts. Specifically, the size of the output for a CIFAR-10 DeepNADE model would be $32 \times 32 \times 3 \times 256 = 786{,}432$ (because each pixel has three channels, and each channel can take on one of 256 different integer values). Therefore, if the input dimension to the final layer was 500 (as it was in the DeepNADE model for binarized-MNIST), the final layer alone would have $500 \times 786{,}432 + 786{,}432 = 394{,}002{,}432$ parameters. While the number of outputs can be reduced for image datasets by using a discretized logistic mixture likelihood (Salimans et al., 2017), this strategy restricts the complexity of the model, and the discretized logistic mixture likelihood is not applicable to datasets where the labels do not have a clear underlying order.

On the other hand, due to the attention mechanism, the DEformer suffers from the same quadratic complexity problem known to plague Transformers. However, recent work in sparse Transformers (e.g., (Child et al., 2019; Zaheer et al., 2020; Beltagy et al., 2020; Kitaev et al., 2020)) may allow the DEformer to scale to larger inputs.

When training DeepNADE, a mask is randomly generated for each sample by: (1) randomly selecting an integer $c \in \{0, \ldots, D-1\}$ to serve as the number of conditioning variables and (2) randomly assigning a value of one to $c$ locations in the mask and assigning a value of zero to the remaining locations. The loss for each sample is then:

$$\mathcal{L} = \frac{D}{D-c} \sum_{d=1}^{D} (1 - m_d)(-\ln(f(\hat{\mathbf{x}}, \mathbf{m})_{(d-1)C + l_d}))$$

where $l_d$ is the label for the $d$-th feature of $\mathbf{x}$, and $\frac{D}{D-c}$ is a scaling factor ensuring the loss for each sample is an unbiased estimator (which is necessary because the error signal is only computed for $D - c$ features of the sample due to the $1 - m_d$ term). In contrast, for the DEformer, there is always an error signal for all of the features of each sample. While MADE (Germain et al., 2015) also produces an error signal for all of the features of each sample, the authors observed that sampling many different weight masks led to the model underfitting, so it is unclear how well MADE can perform fully OADE.

### 5.3. Spatial inputs as feature identities

A number of neural network architectures operate directly on spatial coordinates, which can be interpreted as feature identities in their various contexts (e.g., images (Ha, 2016), point clouds (Guo et al., 2020), and 3D scenes (Sitzmann et al., 2019)). Additionally, Liu et al. (2018) observed that adding channels to feature maps that contain the spatial coordinates of the pixels greatly improved the performance of convolutional neural networks on certain spatial reasoning tasks. However, none of these models are performing autoregressive distribution estimation, nor do they employ the interleaved input design of the DEformer.

## 6. Conclusion

In this paper, we described an alternative approach to OADE where the identities of features are included *alongside* their values in the input. We believe the performance of the DEformer on the binarized-MNIST and POWER datasets is encouraging, and we are excited to see how this architecture can be applied in different contexts.

## Author Contributions

MAA conceived and implemented the architecture, designed and ran the experiments, and wrote the manuscript. AN partially funded MAA and provided the GPUs for the experiments.

## Acknowledgements

We would like to thank Iain Murray and Rasool Fakoor for their helpful feedback.

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
