# OpenReview forum: "The DEformer: An Order-Agnostic Distribution Estimating Transformer"
_ICML.cc/2021/Workshop/INNF — INNF+ 2021 poster_

### Official Review · Reviewer_9p4k · 2021-06-11

**Rating:** Borderline Accept
**Confidence:** 4

**Summary:**

This paper explores autoregressive transformer models for estimation of arbitrary marginal distributions. The model consists of a general transformer architecture that takes pixels and their identifiers (row and column indices in their experiment), applies masking of random ordering, and outputs conditional distributions over the output.

**Justification For Rating:**

The paper is well written, and the topic is a good fit for this workshop.

Calling this "order agnostic" is a bit confusing. The authors write "we shuffled the order of the pixels in each training image to encourage the DEformer to learn a joint distribution of the dataset that is approximately permutation invariant with respect to the ordering of the pixels." While it is true that the model will hopefully learn a density that is approximately invariant to the order in which the pixels are presented, it will definitely not be invariant to permutations of pixels that change their spatial location, since this location is used as input to the network. This is in contrast to the NADE and MADE models that the authors compare against: these models are actually permutation invariant and do not use the spatial structure of the image at all.

The novelty of the presented method is somewhat limited, as models of this type have previously been explored: e.g. in XLNet, which the authors do cite. The experimental validation is also somewhat preliminary (binarized MNIST).

---

### Official Review · Reviewer_LFQV · 2021-06-12

**Rating:** Borderline Accept
**Confidence:** 4

**Summary:**

The paper proposes a transformer-based density model that can represent arbitrary factorization of a joint distribution of data. More specifically, the paper aims at modeling autoregressive models for arbitrary index orderings, but with a single transformer. For a single factorization, the transformer conditions on the order of indices of random variables; for image data, the author uses pixel's xy-coordinate to represent the index. The author emphasizes that the proposed conditioning is computationally more efficient than previous MADE-based models.

During training, the transformer learns arbitrary factorizations by randomizing the orderings; however, one may note that the densities for different index orders won't be consistent with each other. For the test, the learned model computes the likelihood by conditioning on an ordering depending on tasks. As an example, for a given inference task, one can choose an order that is the best fit for the task. Moreover, the model can average an ensemble of the likelihoods of different orderings.

In the experiments, the paper demonstrates that the ensemble estimate of the proposed method achieves as good likelihood as strong autoregressive model baselines. Furthermore, the author shows a potential usage of the proposed method for out-of-distribution detection.

**Justification For Rating:**

Overall I like the general direction of the paper. I found that experiments can be further improved so that the paper can clearly motivate the benefits of learning arbitrary orderings.

---

### Decision · Program_Chairs · 2021-06-14

Accept (poster)